# Attentive State-Space Modeling of Disease Progression

**Ahmed M. Alaa**
ECE Department
UCLA
ahmedmalaa@ucla.edu

**Mihaela van der Schaar**
UCLA, University of Cambridge, and
Alan Turing Institute
{mv472@cam.ac.uk,mihaela@ee.ucla.edu}

## Abstract

Models of disease progression are instrumental for *predicting* patient outcomes and *understanding* disease dynamics. Existing models provide the patient with pragmatic (supervised) predictions of risk, but do not provide the clinician with intelligible (unsupervised) representations of disease pathology. In this paper, we develop the *attentive state-space* model, a deep probabilistic model that learns accurate and interpretable structured representations for disease trajectories. Unlike Markovian state-space models, in which state dynamics are memoryless, our model uses an attention mechanism to create "memoryful" dynamics, whereby attention weights determine the dependence of future disease states on past medical history. To learn the model parameters from medical records, we develop an inference algorithm that jointly learns a compiled inference network and the model parameters, leveraging the attentive representation to construct a variational approximation of the posterior state distribution. Experiments on data from the UK Cystic Fibrosis registry show that our model demonstrates superior predictive accuracy, in addition to providing insights into disease progression dynamic.

## 1 Introduction

Chronic diseases — such as cardiovascular disease, cancer and diabetes — progress slowly throughout a patient's lifetime, causing increasing burden to the patients, their carers, and the healthcare delivery system [1]. The advent of modern electronic health records (EHR) provides an opportunity for building models of disease progression that can *predict* individual-level disease trajectories, and distill *intelligible* and *actionable* representations of disease dynamics [2]. Models that are both highly accurate and capable of extracting knowledge from data are important for informing practice guidelines and identifying the patients' needs and interactions with health services [3, 4, 5, 6].

In this paper, we develop a deep probabilistic model of disease progression that capitalizes on both the interpretable structured representations of probabilistic models and the predictive strength of deep learning methods. Our model uses a state-space representation to segment a patient's disease trajectory into "stages" of progression that manifest through clinical observations. But unlike conventional state-space models, which are predominantly Markovian, our model uses recurrent neural networks (RNN) to capture more complex state dynamics. The proposed model learns hidden disease states from observational data in an unsupervised fashion, and hence it is suitable for EHR data where a patient's record is seldom annotated with "labels" indicating their true health state [7].

Our model uses an *attention* mechanism to capture state dynamics, hence we call it an *attentive state-space* model. The attention mechanism observes the patient's clinical history, and maps it to attention weights that determine how much influence previous disease states have on future state transitions. In that sense, attention weights generated for an individual patient explain the causative and associative relationships between the hidden disease states and the past clinical events for that patient. Because

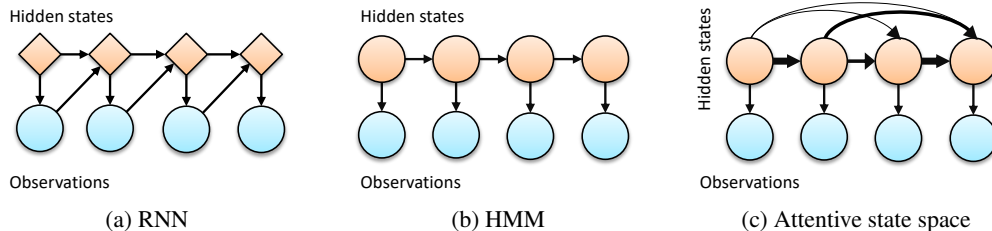

Figure 1: **Sequential data models.** (a) Graphical model for an RNN. $\Diamond$ denotes a deterministic representation, (b) Graphical model for an HMM. $\bigcirc$ denotes probabilistic states, (c) Graphical depiction of an attentive state space model. With a slight abuse of graphical model notation, thickness of arrows reflect attention weights.

the attention mechanism can be made arbitrarily complex, our model can capture complex dynamics while maintaining its structural interpretability. We implement this dynamic attention mechanism via a sequence-to-sequence RNN architecture [8].

Because our model is non-Markovian, inference of posterior disease states is intractable and cannot be conducted using standard forward-backward routines (e.g., [9, 10, 11]). To address this issue, we devise a structured inference network trained to predict posterior state distributions by mimicking the attentive structure of our model. The inference network shares attention weights with the generative model, and uses those weights to create summary statistics needed for posterior state inference. We jointly train the inference and model networks using stochastic gradient descent.

To demonstrate the practical significance of the attentive state-space model, we use it to model the progression trajectories of breast cancer using data from the UK Cystic Fibrosis registry. Our experiments show that attentive state-space models can extract clinically meaningful representations of disease progression while maintaining superior predictive accuracy for future outcomes.

**Related work.** Various predictive models based on RNNs have been recently developed for healthcare settings — e.g., "Doctor AI" [12], "L2D" [13], and "Disease-Atlas" [14]. Unfortunately, RNNs are of a "black-box" nature since their hidden states do not correspond to clinically meaningful variables (Figure 1a). Thus, all the aforementioned methods do not provide an intelligible model of disease progression, but are rather limited to predicting a target outcome.

There have been various attempts to create interpretable RNN-based predictive models using attention. The models in [15, 16, 17] use a reverse-time attention mechanism to learn visit-level attention weights that explain the predictions of an RNN. The main difference between the way attention is used in our model and the way it is used in models like "RETAIN" [15] is that our model applies attention to the latent *state space*, whereas RETAIN applies attention to the observable *sample space*. Hence, attention gives different types of explanations in the two models. In our model, attention interprets the hidden disease dynamics, hence it provides an explanation for the mechanisms underlying disease progression. On the contrary, RETAIN uses attention to measure feature importance, hence it can only explain discriminative predictions, but not the underlying generative disease dynamics.

Almost all existing models of disease progression are based on variants of the HMM model [18, 9, 19]. Disease dynamics in such models are very easily interpretable as they can be perfectly summarized through a single matrix of probabilities that describes transition rates among disease states. Markovian dynamics also simplify inference because the model likelihood factorizes in a way that makes efficient forward and backward message passing possible. However, memoryless Markov models assume that a patient's current state $d$-separates her future trajectory from her clinical history (Figure 1b). This renders HMM-based models incapable of properly explaining the heterogeneity in the patients' progression trajectories, which often results from their varying clinical histories or the chronologies (timing and order) of their clinical events [5]. This limitation is crucial in complex chronic diseases that are accompanied with multiple morbidities. Our model addresses this limitation by creating memoryful state transitions that depend on the patient's entire clinical history (Figure 1c).

Most existing works on deep probabilistic models have focused on developing structured inference algorithms for deep Markov models and their variants [20, 10, 21, 22, 23]. All such models use neural networks to model the transition and emission distributions, but are limited to Markovian dynamics. Other works develop stochastic versions of RNNs for the sake of generative modeling; examples include variational RNNs [24], SRNN [25], and STORN [26]. These models augment stochastic layers to an RNN in order to enrich its output distribution. However, transition and

emission distributions in such models cannot be decoupled, and hence their latent states do not map to clinically meaningful identification of disease states. To the best of our knowledge, ours is the first deep probabilistic model that provides clinically meaningful latent representations, with non-Markovian state dynamics that can be made arbitrarily complex while remaining interpretable.

## 2 Attentive State-Space Models

### 2.1 Structure of the EHR Data

A patient's EHR record, denoted as $\vec{\boldsymbol{x}}_T$, is a collection of sequential follow-up data gathered during repeated hospital visits. We represent a given patient's record as

$$\vec{\boldsymbol{x}}_T = \{x_t\}_{t=1}^T, \tag{1}$$

where $x_t$ is the follow-up data collected during the $t^{th}$ hospital visit, and $T$ is the total number of visits. The follow-up data $x_t \in \mathcal{X}$ is a multi-dimensional vector that comprises information on biomarkers and clinical events, such as treatments and ICD-10 diagnosis codes [2].

### 2.2 Attentive State-Space Representation

We model the progression trajectory of the target disease via a state-space representation. That is, at each time step $t$, the patient's health is characterized by a state $z_t \in \mathcal{Z}$ which manifests through the follow-up data $x_t$. The state space is the (discrete) set of all possible stages of disease progression $\mathcal{Z} = \{1, \ldots, K\}$. In general, progression stages correspond to distinct disease phenotypes. For instance, chronic kidney disease progresses through 5 stages (Stage I to Stage IV), each of which corresponds to a different level of renal dysfunction [27]. We assume that $\{z_t\}_t$ is *hidden*, i.e., the true health state of a patient is not observed, and should be learned in an unsupervised fashion. We model the joint distribution of states and observations via the following factorization:

$$p_\theta(\vec{\boldsymbol{x}}_T, \vec{\boldsymbol{z}}_T) = \prod_{t=1}^T \underbrace{p_\theta(x_t \,|\, z_t)}_{\textbf{Emission}} \underbrace{p_\theta(z_t \,|\, \vec{\boldsymbol{x}}_{t-1}, \vec{\boldsymbol{z}}_{t-1})}_{\textbf{Transition}}, \tag{2}$$

where $\vec{\boldsymbol{z}}_t = \{z_1, \ldots, z_t\}$, $1 \le t \le T$, and $\theta$ is the set of parameters of our model.

**Attentive state transitions.** What makes the model in (2) differ from standard state-space models? The main difference is that the transition probability in (2) assumes that the patient's health state at time $t$ depends on their entire history $(\vec{\boldsymbol{x}}_{t-1}, \vec{\boldsymbol{z}}_{t-1})$. This is a major departure from the standard Markovian assumption, which posits that $p_\theta(z_t \,|\, \vec{\boldsymbol{x}}_{t-1}, \vec{\boldsymbol{z}}_{t-1}) = p_\theta(z_t \,|\, z_{t-1})$, i.e., future states depend only on current state. Most existing disease progression models are Markovian (e.g., [9, 18]).

To capture non-Markovian dynamics, we model the state transition distribution as follows:

$$p_\theta(z_t \,|\, \vec{\boldsymbol{x}}_{t-1}, \vec{\boldsymbol{z}}_{t-1}) = p_\theta(z_t \,|\, \vec{\boldsymbol{\alpha}}_t, \vec{\boldsymbol{z}}_{t-1}), \tag{3}$$

where $\vec{\boldsymbol{\alpha}}_t = \{\alpha_1^t, \ldots, \alpha_{t-1}^t\}$, $\alpha_i^t \in [0, 1], \forall i, \sum_i \alpha_i^t = 1$, is a set of *attention weights* that act as sufficient statistics of future states. The attention weights admit to a simple interpretation: they determine the influences of past state realizations on future state transitions via the linear dynamic

$$p_\theta(z_t \,|\, \vec{\boldsymbol{\alpha}}_t, \vec{\boldsymbol{z}}_{t-1}) = \sum_{t'=1}^{t-1} \alpha_{t'}^{t-1} \, \boldsymbol{P}(z_{t'}, z_t), \, \forall t \ge 1, \tag{4}$$

where $\boldsymbol{P}$ is a baseline state transition matrix, i.e., $\boldsymbol{P} = p_{ij} \in [0, 1]$, $\sum_j p_{ij} = 1$, and the initial state distribution is $\boldsymbol{\pi} = [\, p_1, \ldots, p_K \,]$. The attention weights $\vec{\boldsymbol{\alpha}}_t$ assigned to all previous state realizations at time $t$ are generated using the patient's *context* $\vec{\boldsymbol{x}}_t$ via an attention mechanism $A$ as follows:

$$\vec{\boldsymbol{\alpha}}_t = A_t(\vec{\boldsymbol{x}}_t). \tag{5}$$

where $A$ is a deterministic algorithm that generates a sequence of functions $\{A_t\}_t$, $A_t : \mathcal{X}^t \to [0, 1]^t$. We specify our choice of the attention mechanism in Section 2.3.

**Emission distribution.** The follow-up data $x_t = (x_t^c, x_t^b)$ comprises both a continuous component $x_t^c$ (e.g., biomarkers and test results) and a binary component $x_t^b$ (e.g., clinical events and ICD-10

codes). To capture both components, we model the emission distribution in (2) through the following factors $p_\theta(x_t \mid z_t) = p_\theta(x_t^b \mid x_t^c, z_t) \cdot p_\theta(x_t^c \mid z_t)$, where

$$p_\theta(x_t^c \mid z_t) = \mathcal{N}(\mu_{z_t}, \Sigma_{z_t}), \quad p_\theta(x_t^b \mid x_t^c, z_t) = \text{Bernoulli}(\text{Logistic}(x_t^c, \Lambda_{z_t})). \tag{6}$$

The model in (6) specifies a state-specific distribution for binary (Bernoulli) and continuous (Gaussian) variables, with state-specific emission distribution parameters $(\mu_{z_t}, \Sigma_{z_t}, \Lambda_{z_t})$. This, an attentive state-space model can be completely specified through the parameter set $\theta = (\pi, P, A, \mu, \Sigma, \Lambda)$.

**Generality of the attentive representation.** For particular choices of the attention mechanism in (4), our model reduces to various classical models of sequential data as shown in Table 1.

The generality of the attentive state representation is a powerful feature because it implies that by learning the attention functions $\{A_t\}_t$, we are effectively testing the structural assumptions of various commonly-used time series models in a data-driven fashion.

| Model | Attention mechanism |
|:---:|:---:|
| **HMM [9]** | $\alpha_{t-1}^t = 1, \alpha_j^t = 0, j \leq t-2.$ |
| **Order-$m$ HMM [28]** | $\alpha_j^t = \mathbf{1}_{\{m \leq j \leq t-1\}}, j \leq t-2.$ |
| **Variable-order HMM [29]** | $\alpha_j^t \in \{0, \bar{n}^{-1}\}, \bar{n} = \sum_i \mathbf{1}_{\{\alpha_i^t \geq \gamma\}}.$ |

Table 1: Representation of familiar elementary functions in terms of.

### 2.3 Sequence-to-sequence Attention Mechanism

To complete the specification of our model, we now specify the attention mechanism $A$ in (5). Recall that $A$ is a sequence of deterministic functions that map a patient's context $\vec{x}_t$ to a set of attention weights $\vec{\alpha}_t$ at each time step. Since our model must output an entire sequence of attention weights every time step, we implement $A$ via a sequence-to-sequence (Seq2Seq) model [8].

Our Seq2Seq model uses LSTM encoder-decoder architecture as shown in Figure 2. For each time step $t$, the patient context $\vec{x}_t$ is fed to the LSTM encoder, and the final state of the encoder, $h_t$, is viewed as a fixed-size representation of the patient's context, and is passed together with the last output $O$ to the decoder side.

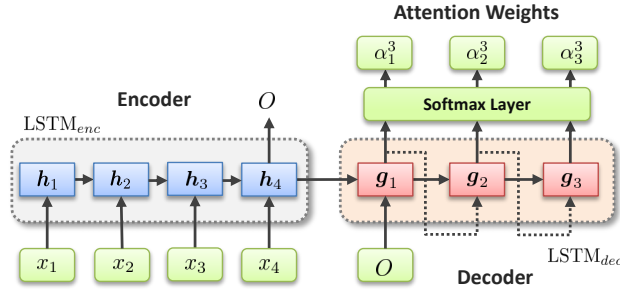

Figure 2: Seq2Seq architecture for the attention mechanism $A$.

In the decoding phase, the last state of the encoding LSTM is used as an initial state of the decoding LSTM, and $O$ is used as its first input. Then, the decoding LSTM iteratively uses its output at one time step as its input for the next step. After $t-1$ decoding iterations, we collect the $t-1$ (normalized) attention weights via a Softmax output layer.

The main difference between our architecture and other Seq2Seq models — often used in language translation tasks [30, 8] — is that in our case, we learn an entire sequence of attention weights for each of the $T$ data vectors in $\vec{x}_T$. We achieve this by running $t-1$ decoding iterations to collect $t-1$ outputs for every single encoding step. Moreover, in our setup attention sequence is the target sequence being learned. This should not be confused with other Seq2Seq schemes with attention, where attention is used as an intermediate representation within a decoding procedure [31].

### 2.4 Why Attentive State Space Modeling?

Most existing models of disease progression are based on Hidden Markov models [19, 9, 18, 32]. However, the Markovian dynamic is oversimplified: in reality, a patient transition to a given state depends not only on her current stage, but also on her individual history of past clinical events [1]. In this sense, a Markov models is of a "one-size-fits-all" nature — under a Markov model, all patients at the same stage of progression would have the same expected future trajectory, irrespective of their potentially different individual clinical histories. Because Markov models explain away individual-level variations in progression trajectories, their interpretable nature should be thought

of as a bug and not a feature, i.e., a Markov model is easily interpretable only because it *does not explain much*, it only encodes our own prior assumptions about disease dynamics.

Attentive state space models overcome the shortcomings of Markov models by using attention weights to create non-stationary, variable-order generalization of Markovian transitions, whereby the dynamics of each patient changes over time based on her *individual* clinical context. An attentive state model can learn state dynamics that are as complex as those of an RNN, but through the factorization in (2), it ensures that its hidden states correspond to clinically meaningful disease states.

## 3 Attentive Variational Inference

Learning the model parameter $\theta$ and inferring a patient's health state in real-time requires computing the posterior $p_\theta(\vec{z}_t \,|\, \vec{x}_t)$. However, the non-Markovian nature of our model renders posterior computation intractable. In this Section, we develop a variational learning algorithm that jointly learns the model parameter $\theta$ and a structured inference network that approximates the posterior $p_\theta(\vec{z}_t \,|\, \vec{x}_t)$. We show that the attentive representation proposed in Section 2 is useful not only for improving predictions and extracting clinical knowledge, but also can help improve structured inference.

### 3.1 Variational Lower Bound

In variational learning, we maximize an evidence lower bound (ELBO) for the data likelihood, i.e.,

$$\log p_\theta(\vec{x}_T) \geq \mathbb{E}_{q_\phi}\left[\log p_\theta(\vec{x}_T, \vec{z}_T) - \log q_\phi(\vec{z}_T \,|\, \vec{x}_T)\right],$$

where $q_\phi(\vec{z}_T \,|\, \vec{x}_T)$ is a variational distribution that approximates the posterior $p_\theta(\vec{z}_T \,|\, \vec{x}_T)$. We model the variational distribution $q_\phi(\vec{z}_T \,|\, \vec{x}_T)$ using an *inference network* that is trained jointly with the model through the following optimization problem [33, 34]:

$$\theta^*, \phi^* = \arg\max_{\theta,\phi} \mathbb{E}_{q_\phi}\left[\log p_\theta(\vec{x}_T, \vec{z}_T) - \log q_\phi(\vec{z}_T \,|\, \vec{x}_T)\right]. \tag{7}$$

By estimating $\theta$ and $\phi$ from the EHR data, we recover the generative model $p_\theta(\vec{x}_T, \vec{z}_T)$, through which we can extract clinical knowledge, and the inference network $q_\phi(\vec{z}_T \,|\, \vec{x}_T)$, through which we can use to infer the health trajectory of the patient at hand.

### 3.2 Attentive Inference Network

We construct the inference network $q_\phi(\vec{z}_T \,|\, \vec{x}_T)$ so that it mimics the structure of the true posterior [20]. Recall that the posterior factorizes as follows:

$$p_\theta(\vec{z}_T \,|\, \vec{x}_T) = p_\theta(z_1 \,|\, \vec{x}_T) \prod_{t=2}^{T} p_\theta(z_t \,|\, \vec{\alpha}_{t-1}, \vec{z}_{t-1}, \vec{x}_{t:T}).$$

Consequently, we impose a similar factorization on the inference network, i.e.,

$$q_\phi(\vec{z}_T \,|\, \vec{x}_T) = q_\phi(z_1 \,|\, \vec{x}_T) \prod_{t=2}^{T} q_\phi(z_t \,|\, \vec{\alpha}_{t-1}, \vec{z}_{t-1}, \vec{x}_{t:T}). \tag{8}$$

To capture the factorization in (8), we use the architecture in Figure 3 to construct an inference network that mimics the attentive structure of the generative model. In this architecture, a "combiner function" $C(.)$ is fed with all the sufficient statistics of a state $z_t$, and outputs its posterior distribution. The combiner uses the attention weights created by $A$ to condense summary statistics of $z_t$.

As dictated by (8), the parent nodes of $z_t$ are the attention weights $\vec{\alpha}_t$, the previous states $\vec{z}_{t-1}$ and the future observations $\vec{x}_{t:T}$. The inference network encodes these sufficient statistics as follows. The attention weights $\vec{\alpha}_t$ are shared with the attention network in Figure 2. The future observations $\vec{x}_{t:T}$ are summarized at time $t$ via a backward LSTM that reads $\vec{x}_T$ in a reversed order as shown in Figure 3. Finally, the previous states $\vec{z}_{t-1}$ are sampled from the combiner functions at previous time steps as described below.

**Posterior sampling.** In order to sample posterior state trajectories from the inference network, we iterate over the combiner function $C(.)$ for $t \in \{1, \ldots, T\}$ as follows:

$$\tilde{p}_t = C(h_q^t, \vec{\alpha}_t, (\tilde{z}_1, \ldots, \tilde{z}_{t-1}) \,|\, \pi, P),$$
$$\tilde{z}_t \sim \text{Multinomial}(\tilde{p}_t), \qquad (9)$$

where $\tilde{p}_t = (\tilde{p}_1^t, \ldots, \tilde{p}_K^t), \sum_k \tilde{p}_k^t = 1$, is the posterior state distribution estimated by the inference network at time $t$, and $h_q^t$ is the $t^{th}$ state of the backward LSTM in Figure 3, which summarizes the information in $\vec{x}_{t:T}$. As we can see in (9), at each time step $t$, the combiner function takes as an input *all* the previous states $(\tilde{z}_1, \ldots, \tilde{z}_{t-1})$ sampled by earlier executions of the combiner function. The dashed blue lines in Figure 3 depict the passage of older state samples to later executions of the combiner function.

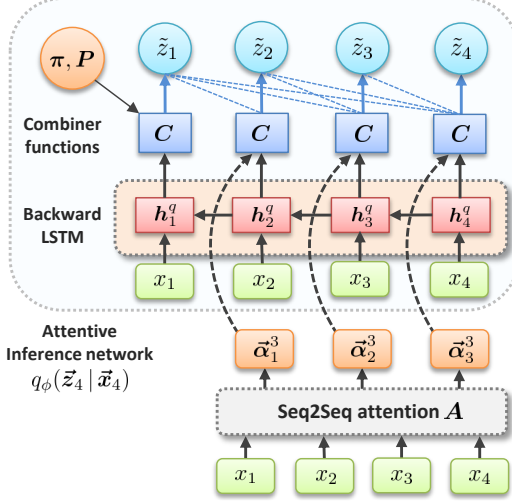

Figure 3: Attentive inference network.

The combiner function estimates the posterior $\tilde{p}_t$ by emulating the state transition model in (4), i.e.,

$$\tilde{p}_{k,forward}^t = \sum_{t'=1}^{t-1} \alpha_{t'}^t P(\tilde{z}_{t'}, k), \; k \in \{1, \ldots, K\},$$
$$\tilde{h}_q^t = [h_q^t, \tilde{p}_{1,forward}^t \cdots, \tilde{p}_{K,forward}^t],$$
$$\tilde{p}_t = \text{Softmax}(W_q^\top \tilde{h}_q^t + b_q). \qquad (10)$$

As shown in (10), the combiner emulates the generative model to compute an estimate of the "filtering" distribution $\tilde{p}_{k,forward}^t \approx p_\theta(z_t \,|\, \vec{x}_t)$, i.e., it attends to previously sampled states with proportions determined by the attention weights. Then, to augment information from the future observations $\vec{x}_{t:T}$, it concatenates the filtering distribution with the backward LSTM state and estimates the posterior through a Softmax output layer.

### 3.3 Learning with Stochastic Gradient Descent

In order to simultaneously learn the parameters of the generative model and inference network, we use stochastic gradient descent to solve (7) as follows:

1. Sample $(\tilde{z}_1^{(i)}, \ldots, \tilde{z}_T^{(i)}) \sim q_\phi(\vec{z}_T \,|\, \vec{x}_T)$, $i = 1, \ldots, N$.
2. Estimate ELBO $\hat{\mathcal{L}} = \frac{1}{N} \sum_i \ell_{\theta,\phi}(\vec{x}_T, \tilde{z}_1^{(i)}, \ldots, \tilde{z}_T^{(i)})$.
3. Estimate the gradients $\nabla_\theta \hat{\mathcal{L}}$ and $\nabla_\phi \hat{\mathcal{L}}$.
4. Update $\phi$ and $\theta$.

In Step 2, the term $\ell_{\theta,\phi}(.)$ denotes the objective function in (7). We estimate the gradients in Step 3 via stochastic backpropagation [35]. In Step 4, we use ADAM [36] to update the parameters of the attention mechanism (Figure 2) and the inference network (Figure 3). The emission parameters are updated straightforwardly by their maximum likelihood estimates.

**Rao-Blackwellization via attention.** As we have seen, our attentive inference network architecture enables sharing parameters between the generative model and the inference model, which would definitely accelerate learning. Another key advantage of the attentive structure $q_\phi(z_t \,|\, \vec{x}_T)$ is that it acts as a Rao-Blackwellization of the conventional structured inference network which conditions on *all* observation (i.e., $q_\phi(z_t \,|\, \vec{x}_T)$ [20, 11, 21]). Because attention weights (together with $\vec{z}_{t-1}$ and $\vec{x}_{t:T}$)) act as sufficient statistics for state transitions, our inference networks guides the posterior to focus only on the pieces of information that matter. Rao-Blackwellization helps reduce the variance of gradient estimates (Step 3 in the learning algorithm above), and hence accelerate learning [37].

## 4 Experiments

In this Section, we use our attentive state-space framework to model cystic fibrosis (CF) progression trajectories. CF is a life-shortening disease that causes lung dysfunction, and is the most common genetic disease in Caucasian populations [38]. Experimental details are listed hereunder.

**Implementation.** We implemented our model using `Tensorflow`[1]. The LSTM cells in both the attention network (Figure 2) and the inference network (Figure 3) had 2 hidden layers of size 100. The model and inference networks were trained using ADAM with a learning rate of $5 \times 10^{-4}$, and a mini-batch size of 100. The same hyperparameters' setting was used for all baseline models involving RNNs. All prediction results reported in this Section where obtained via 5-fold cross-validation.

**Data description.** We used data from a cohort of patients enrolled in the UK CF registry, a database held by the UK CF trust[2]. The dataset records annual follow-ups for 10,263 patients over the period from 2008 and 2015, with a total of 60,218 hospital visits. Each patient is associated with 90 variables, including information on 36 possible treatments, diagnoses for 31 possible comorbidities and 16 possible infections, in addition to biomarkers and demographic information. The FEV1 biomarker (a measure of lung function) is the main measure of illness severity in CF patients [39].

**Training.** In Figure 4, we show the model's log-likelihood (LL) versus the number of training epochs. As we can see, the more training iterations we apply, the better the model likelihood gets: it jumped from $-4 \times 10^{-6}$ in the initial iterations to $-8 \times 10^{-5}$ after training was completed. The best value of the log-likelihood is 0, which is achieved when the inference network $q_\phi(z_t \,|\, \vec{\boldsymbol{x}}_T)$ coincides with the true model $p_\theta(z_t \,|\, \vec{\boldsymbol{x}}_T)$, and the observed data likelihood given the model is 1. Attentive inference is accurate because it utilizes the minimally sufficient set if past information, which reduces the variance in gradient estimates (Section 3.3).

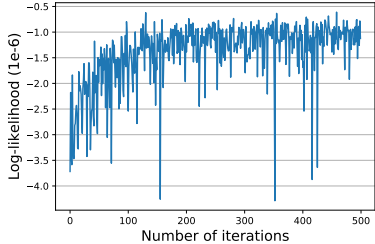

Figure 4: LL vs. training epochs.

**Use cases.** We assess our model with respect to the two use cases it was designed for: (1) extracting clinical knowledge on disease progression mechanisms from the data, and (2) predicting a patient's health trajectory over time. We assess each use case separately in Sections 4.1 and 4.2.

### 4.1 Understanding CF Progression Mechanisms

**Population-level phenotyping.** Our model learned a representation of $K = 3$ CF progression stages (Stages 1, 2 and 3) in an unsupervised fashion, i.e., each stage is a realization of the hidden state $z_t$.

As we show in what follows, each learned progression stage corresponded to a clinically distinguishable phenotype of disease activity. The learned baseline transition matrix was

$$\boldsymbol{P} = \begin{bmatrix} 0.85 & 0.10 & 0.05 \\ 0.13 & 0.72 & 0.15 \\ 0.24 & 0.10 & 0.66 \end{bmatrix}.$$

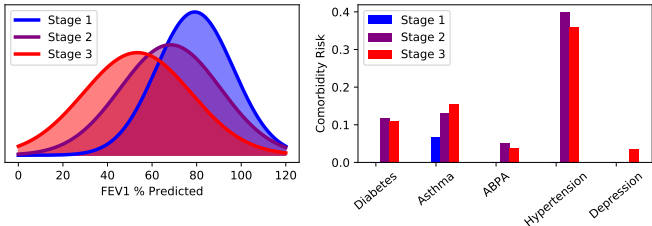

Figure 5: Distribution of observations in each progression stage.

The FEV1 biomarker is currently used by clinicians as a proximal measure of a patient's health in order to guide clinical and therapeutic decisions [40]. In order to check that the learned progression stages correspond to different levels of disease severity, we plot the estimated mean of the emission distribution for the FEV1 biomarker in Stages 1, 2 and 3 in Figure 5 (left). As we can see from Figure 5 (left), the mean values of the FEV1 biomarker in each stage were 79%, 68% and 53%, respectively. These values matched with the cutoff values on FEV1 used in current guidelines for referring critically-ill patients to a lung transplant [40]. Thus, the learned progression stages can be translated into actionable information for clinical decision-making.

---

| Model | Diabetes AUC-ROC | ABPA AUC-ROC | Depression AUC-ROC | Pancreatitus AUC-ROC | P. Aeruginosa AUC-ROC |
|---|---|---|---|---|---|
| **Attentive SS** | **0.709 ± 0.02** | **0.787 ± 0.01** | **0.751 ± 0.03** | **0.696 ± 0.04** | **0.680 ± 0.01** |
| **HMM** | 0.625 ± 0.02 | 0.686 ± 0.03 | 0.667 ± 0.08 | 0.625 ± 0.04 | 0.610 ± 0.02 |
| **RNN** | 0.634 ± 0.03 | 0.727 ± 0.10 | 0.575 ± 0.01 | 0.590 ± 0.06 | 0.654 ± 0.01 |
| **LSTM** | 0.675 ± 0.03 | 0.740 ± 0.07 | 0.609 ± 0.12 | 0.578 ± 0.05 | 0.671 ± 0.01 |
| **RETAIN** | 0.610 ± 0.06 | 0.718 ± 0.05 | 0.580 ± 0.09 | 0.600 ± 0.08 | 0.676 ± 0.02 |

Table 2: Performance of the different competing models for the 5 prognostic tasks under consideration.

The progression stages learned by our model represented clinically distinguishable phenotypes with respect to multiple clinical variables. To illustrate these phenotypes, in Figure 5 (right) we plot the risks of various comorbities (Diabetes, asthma, ABPA, hypertension and depression) for patients in the 3 CF progression stages learned by the model. (Those risks were obtained directly from the learned emission distribution corresponding to the binary component $x_t^b$ of the clinical observation $x_t$.) As we can see, the incidences of those comorbidities and infections increase significantly in the more severe progression Stages 2 and 3 as compared to Stage 1.

**Individualized contextual diagnosis.** Population level modeling of disease stages can be already obtained with simple HMM models, but our model captures more complex dynamics that are specific to individuals, and can be made non-Markovian and non-stationary depending on the patient's context. To demonstrate the complex and non-stationary nature of the learned state dynamics, we plot the average attention weights assigned to the patients' previous state realizations in every "chronological" time step of a patient trajectory. The average attention weights per time step is plotted in Figure 6.

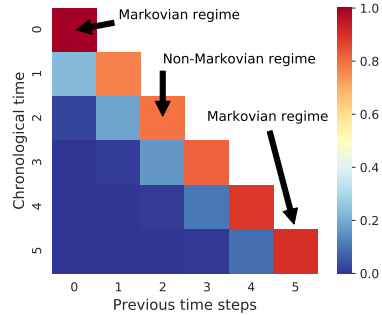

Figure 6: Average attention weights over time.

As we can see, a patient's state trajectory behaves in a quasi-Markovian fashion (only current state takes all the weight) only on its edges. That is, at the first time step and the last time step, the only thing that matters for prediction is the patient's current state. This is because in the first time step, the patient has no history, whereas in the final step, the patient is already in the most severe state and hence her current health deterioration depends overrides all past clinical events. Memory becomes important only in intermediate Stages — this is because patients in Stages 2 and 3 are more likely to have been diagnosed with more comorbidities in the past.

## 4.2 Predicting Prognosis

As we have seen in Section 4.2, our model is capable of extracting clinical intelligence from data, but does this compromise its predictive ability? To test the predictive ability of attentive state-space models, we sequentially predict the 1-year risk of 4 comorbidities (ABPA, diabetes, depression and pancreatitus), and 1 lung infections (Pseudomonas Aeruginosa) that are common in the CF population. We use the area under receiver operating characteristic curve (AUC-ROC) for performance evaluation. We report average AUC-ROC with 95% confidence intervals. We compare our model with 4 baselines: a vanilla RNN and an LSTM trained for sequence prediction, a state-of-the-art predictive model for healthcare data known as RETAIN [17, 15], and an HMM.

As we can see in Table 2, our model did not incur any performance loss when compared to models trained and tailored for the given prediction task (RNN, LSTM and RETAIN), and was in fact more accurate on all of the 5 tasks. The source of the predictive power in attentive state-space models comes from the usage of LSTM networks to model state dynamics in a low-dimensional space that summarizes the 90 variables associated with each patient. While HMMs can also learn interpretable representations of disease progression, they displayed modest predictive performance because of their oversimplified Markovian dynamics. Because attentive state-space models are capable of combining the interpretational benefits of probabilistic models and the predictive strength of deep learning, we envision them being used for large-scale disease phenotyping and clinical decision-making.

## Acknowledgments

This work was supported by the National Science Foundation (NSF grants 1462245 and 1533983), and the US Office of Naval Research (ONR). The data for our experiments was provided by the UK Cystic Fibrosis Trust. We thank Dr. Janet Allen (Director of Strategic Innovation, UK Cystic Fibrosis Trust) for the vision and encouragement. We thank Rebecca Cosgriff and Elaine Gunn for the help with data access, extraction and analysis.

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
