[Reviews · NeurIPS 2019]

Reviewer 1



Originality: The paper is innovative. Disease progression modelling is a very timely application, and not many papers that present experimental and clinically meaningful machine learning results exist. Clarity: The paper is very clearly written. It was my pleasure to read and review it. As complimented above, the authors have excelled in conceptualising a problem, composing it as a machine learning task, and writing and referencing the related report in a way that is meaningful and engaging for both the computer science/engineering and health care/sciences communities. It is very rare to see a paper that delivers so well to such a diverse audience. Also findings are presented and visualised (as Tables and Figures) in a mature, meaningful, and comparative way. Quality: Although many of the methods are perhaps unsurprising and perhaps not as inspiring to machine learning method developers, they are well justified, compared, and related. The contribution of this paper really is that it puts machine learning into practice in a way that is meaningful to clinicians. Significance: The authors consider a societally significant problem of modelling disease progression in way that is clinically meaningful by providing actionable insights to inform their clinical judgement and decision-making. It is important that studies of this kind are presented in NeurIPS to attract more people to contribute to this important topic of medical informatics.

Reviewer 2



# Originality Discrete state space models are often considered to be interpretable because they essentially cluster the elements of time series data (while accounting for time dependence). The key idea in this paper is to maintain this property of discrete state space models while relaxing the stationary Markov assumption on the transition probabilities that we typically use to simplify inference. Although this idea is not new (e.g. semi-Markov models, or Markov models that augment the state with a larger history), the proposed mechanism for relaxing the assumption does seem to be original. The variational inference algorithm for this model also seems to be new. # Quality [1] I think that the HMM baseline is unnecessarily weak. In practice, we can relax the "strict" Markov assumption (i.e. the state in year $t + 1$ is conditionally independent of the past given the state at year $t$) by augmenting the state with the past $h > 1$ years. This keeps the inference exact and relatively easy to implement. Although the state space can grow quite large, it still may not be quite as big of a burden as fitting a complicated model with poorly understood convergence properties. [2] I also thought that the analysis of the proposed variational inference algorithm could be much stronger. The main theoretical motivation for the VI algorithm is Rao-Blackwellization, but I was confused by this claim. Typically, this means that an estimator (i.e. a function of the data) is replaced with its expected value conditioned on some piece of observable information. The paper doesn't show how this definition connects to the proposed VI algorithm, which weakened the argument. The experimental results did not support this claim either. The only result shows the change in negative log-likelihood as a function of epochs for the proposed algorithm and two reasonable alternatives. Although the proposed VI algorithm does seem to achieve a better upper bound on the NLL, there's no evidence that this is due to the reasons that the authors discuss in Section 3. Moreover, the NLL doesn't answer the question of whether the inferences using the proposed VI algorithm are actually better than those obtained using the baselines. Two inference-related questions I have are: (1) Do these simpler inference algorithms recover similar stages of Cystic Fibrosis? (2) How do the prediction results in Table 2 change when we use these alternative inference algorithms? # Clarity The paper is clearly written, and I enjoyed reading it. # Significance I think the method has the potential to be make an impact on the ML+health field, but I have a few reservations. First, this algorithm might be difficult for practitioners to use in practice. Comparison to a stronger HMM baseline (e.g. the one I described above in "Quality", which is simple to implement) could help to show that the increased difficulty of implementation is justified. Second, when internal details about a model are given to clinicians to help make predictions more "interpretable", I believe that it is important to quantify uncertainty. For instance, it appears that patients in Stages 2 or 3 of CF are at much higher risk of diabetes than those at Stage 1. How stable is this pattern? Would it change if we fit the model using, say, a different random seed? In general, what can we say about the reliability/stability of the latent states/stages that this model learns? # Minor questions: How did you choose the number of states for attentive state space model in the CF analysis? And how did you initialize the states?

Reviewer 3



Overall: I found this paper well-written and convincing. The authors do a strong job justifying the proposed model and positioning it relative to standard models in the field. I have a few questions, but overall, think it is a very strong applied paper. Questions/comments: 1. Clinical visits are generally irregularly spaced and rife with missing measurements. How was this handled in the context of your discrete time model? 2. Beyond evaluation in the context of this paper, why restrict the model to have the same number of states as an existing disease phenotype scheme? In particular, we cannot assume that the unsupervised model will learn back an existing phenotype scheme and, in many cases, it is not clear that we would want to. 3. Line 159: Similarly, we *hope* that the factorization in (2) produces clinically meaningful disease states, but it is by no means a guarantee. 4. Line 177: This posterior factorization was not been previously discussed. I recommend adding a section in the supplementary material that gives more details on its derivation. Maybe even just a simple three time step example derived from the graphical model would suffice. 6. Line 183: \vec{\alpha}_t --> \vec{\alpha}_{t-1} 7. Figure 4: I'm not sure that the training NLL is very informative here since we would expect a more expressive model to fit the training data better regardless of how well it generalizes. I would recommend a held-out NLL plot instead. Also, are the lines for Mean-Field and Attentive actual NLL values or are they the ELBO? 8. Line 237: The learned transition matrix has non-zero probabilities for transitioning backward from more severe stages to less severe ones. Is the existing CF progression monotonic or does it allow for backwards transitions? If so, this seems like the kind of constraint that could be easily encoded in the model.

[Author Response · NeurIPS 2019]

[[ **Reviewer 1** ]] Thank you for your feedback. We will fix the reference formatting in the final manuscript. ∎ **Statistical significance:** In Table 2, we have already included the 95% confidence intervals obtained via stratified cross-validation. In the final manuscript, we will include the $p$-values and confidence intervals for the results in Figures 5 and 6 as well. ∎ **Summary section:** Thank you for this suggestion. To address this comment, we will move Subsection 2.4 (lines 145 - 159) towards the end of Section 3, and add a summary of the inference algorithm to this Subsection.

[[ **Reviewer 2** ]] Thank you for the valuable comments and suggestions. ∎ **Minor questions:** Number of states were selected based on the BIC criterion, and the selected number conformed with existing clinical guidelines (please refer to response "Disease phenotypes" for Reviewer 3). Model parameters were initialized via $K$-means clustering applied to the unrolled time-series data. ∎ **Inference-related questions:** Simpler inference algorithms recover emission distributions similar to our algorithm but do a poorer job in recovering state dynamics (transition probabilities and attention weights). Consequently, the predictive accuracy of mean-field and Markovian inference were better than HMMs but slightly worse than RNNs. Accuracy results for both inference strategies will be added to the supplementary.

∎ **Originality & baselines:** To emphasize the difference between our model and variants of HMMs, we have collected new results for extra baselines including a $4^{th}$-order HMM (4-HMM), a Hidden Semi-Markov model (HSMM), and a factorial HMM (FHMM). Among these, the most competitive was the 4-HMM, but its (predictive) performance was still significantly worse than the RNN benchmark. The reason our model outperforms HMM variants is that it adapts its state dynamics in a non-stationary fashion via time-varying attention weights. The Figure above depicts one patient's trajectory by visualizing the attention weights allocated to previous states as they are being updated each time step. As we can see, the model exhibits Markovian dynamics towards the beginning and the end of the trajectory, and non-Markovian dynamics in

the intermediate steps. Each patient will have their own attention profile (and hence their own contextualized state dynamics) based on their individual clinical observations. This enriches the discriminative power of our model compared to HMM variants that entail stationary/fixed dynamics. ∎ **Rao-Blackwellization:** Given the distributional specification in Section 2 and an RNN estimator $\text{RNN}(\boldsymbol{x}, \boldsymbol{z})$ of the posterior state distribution, it can be shown that our inference network is equivalent to the conditional expectation $\mathbb{E}[\text{RNN}(\boldsymbol{x}, \boldsymbol{z}) \,|\, T(\boldsymbol{x}, \boldsymbol{z})]$ with respect to the *sufficient statistic* $T(\boldsymbol{x}, \boldsymbol{z})$. The sufficient statistic in our model is the attention weight sequence, i.e., $T(\boldsymbol{x}, \boldsymbol{z}) = \boldsymbol{\alpha}$. The proof of this equivalence follows from the seminal work on Rao-Blackwellized particle filters in [R2]; we will provide the corresponding derivation for our model in the supplementary material. As pointed out by the reviewer, Rao-Blackwellized estimators reduce the estimation variance: we will demonstrate that our algorithm minimizes the estimation variance by annotating the cross-validation variance of all estimators in Figure 4. ∎ **Uncertainty:** Reliability/stability of phenotype inferences were judged by the variance of model parameters obtained via 5-fold cross-validation: all findings were statistically significant (refer to response "Statistical significance" for Reviewer 1). Posterior model uncertainty can be easily quantified via test-time Monte Carlo dropout applied to the inference network as in [R1], without any modifications to our algorithm. A fully Bayesian approach with priors over model parameters is also possible. Detailed discussion on uncertainty quantification will be added to the final manuscript.

[[ **Reviewer 3** ]] Thank you for the helpful comments and suggestions. In the final manuscript, we will fix the typo in line 183 (Q6) and add a supplementary Section with details on the factorization in line 177 (Q4). ∎ **Irregularly spaced visits (Q1):** Our framework can be straightforwardly extended to the continuous-time (CT) setup by replacing the discrete-time RNNs in the model and inference networks with CT RNN models such as *phased LSTMs* [R3]. Since phased LSTMs also handle asynchronous observations (See [R3]), the inference network can be trained to update the posterior state distribution based on partial observations at arbitrary time steps, emulating a "memoryful CT HMM". We have already implemented this variant of our model based on phased LSTMs — a related discussion will be added to the supplementary. ∎ **Disease phenotypes (Q2, Q3 and Q8):** The model does not need to be restricted to the number of states for existing phenotype schemes (Q1). In general, model selection criteria — such as the Bayesian information criterion (BIC) — would be used to select the number of states in a data-driven fashion. In our experiment, the number of states selected by BIC matched that of existing phenotypic scheme. As it is the case in all unsupervised setups, the meaningfulness of the states cannot be guaranteed and will have to be judged by experts (Q2). Backward transitions in CF are possible through prescriptions of antibiotics. Monotonic progression (e.g., in Kidney diseases) can be easily encoded in our model by setting the relevant transition parameters to 0 (Q8). ∎ **NNL (Q7):** In the final manuscript, we will replace the in-sample NNL (Figure 4) with the held-out NNL. Note that in all of the 3 cases in Figure 4, the model is fixed and only the inference strategy is changed. The NNL displayed in Figure 4 is the true LL and not the ELBO.

**References:** [R1] Y. Gal, et al. "Dropout as a Bayesian approximation: Insights and applications", ICML, 2015. [R2] A. Doucet, et al. "Rao-Blackwellised particle filtering for dynamic Bayesian networks." UAI, 2000. [R3] D. Neil, et al. "Phased LSTM: Accelerating recurrent network training for long or event-based sequences." NeurIPS, 2016.

[Meta-Review · NeurIPS 2019]

The authors propose a generative discrete state space model and novel variational inference algorithm for modeling disease trajectories. Overall, reviewers found the paper well-written and convincing. However, the authors are encouraged to strongly consider the feedback received. Specifically, in preparing the camera-ready version please incorporate experiments comparing to common extensions of simple well-understood methods (e.g., higher-order HMMs).